# A Histopathological and Surgical Analysis of Gastric Cancer: A Two-Year Experience in a Single Center

**DOI:** 10.3390/cancers17132219

**Published:** 2025-07-02

**Authors:** Cătălin Prodan-Bărbulescu, Flaviu Ionuț Faur, Norberth-Istvan Varga, Rami Hajjar, Paul Pașca, Laura-Andreea Ghenciu, Cătălin Ionuț Vlăduț Feier, Alis Dema, Naomi Fărcuț, Sorin Bolintineanu, Amadeus Dobrescu, Ciprian Duță, Dan Brebu

**Affiliations:** 1Department I, Discipline of Anatomy and Embriology, “Victor Babes” University of Medicine and Pharmacy, 300041 Timisoara, Romania; catalin.prodan-barbulescu@umft.ro (C.P.-B.); s.bolintineanu@umft.ro (S.B.); 22nd Surgery Clinic, Timisoara Emergency County Hospital, 300723 Timisoara, Romania; rami.hajjar@umft.ro (R.H.); paul.pasca@umft.ro (P.P.); dobrescu.amadeus@umft.ro (A.D.); duta.ciprian@umft.ro (C.D.); brebu.dan@umft.ro (D.B.); 3Doctoral School, “Victor Babes” University of Medicine and Pharmacy Timisoara, Eftimie Murgu Square 2, 300041 Timisoara, Romania; 4X Department of General Surgery, “Victor Babes” University of Medicine and Pharmacy, 300041 Timisoara, Romania; 5Researching Future Chirurgie 2 (CHIR 2), Timisoara Emergency County Hospital, 300723 Timisoara, Romania; 6Department III, Discipline of Pathophysiology, “Victor Babes” University of Medicine and Pharmacy, 300041 Timisoara, Romania; bolintineanu.laura@umft.ro; 7Center for Translational Research and Systems Medicine, “Victor Babes” University of Medicine and Pharmacy, 300041 Timisoara, Romania; 8First Discipline of Surgery, Department X-Surgery, “Victor Babes” University of Medicine and Pharmacy, 2 E. Murgu Sq., 300041 Timisoara, Romania; catalin.feier@umft.ro; 9First Surgery Clinic, “Pius Brinzeu” Clinical Emergency Hospital, 300723 Timisoara, Romania; 10Department of Pathology, “Pius Brinzeu” County Clinical Emergency Hospital, 300723 Timisoara, Romania; dema.alis@umft.ro (A.D.); naomi.cociuba@yahoo.ro (N.F.); 11ANAPATMOL Research Center, ‘Victor Babes’ University of Medicine and Pharmacy of Timisoara, 300041 Timisoara, Romania

**Keywords:** gastric neoplasia, histopathological characteristics, curability parameters, WHO classification, Lauren classification, morphopathological analysis, COVID-19, pandemic

## Abstract

Gastric neoplasms are a prevalent malignant pathology with high incidence and prevalence, necessitating precise histopathological characterization for optimal treatment planning. A retrospective analysis of 67 gastric neoplasia subjects from January 2020 to December 2021 revealed that adenocarcinoma was the most common histologic type (91.0%), with most tumors diagnosed as Stage III (40.3%). The intestinal type was the most common (49.2%), followed by diffuse (36.1%) and mixed (14.8%). Poorly differentiated tumors (G3) accounted for 53.7% of cases. The surgical team performed curative resection in 75% of patients, achieving R0 margins in 88% of cases. Subtotal gastrectomy with D2 lymphadenectomy yielded the highest curative success rate (96.6% R0 re-section). Two significant correlations were found between age and tumor grade and between the number of lymph nodes examined and the number of lymph nodes invaded. Adenocarcinomas showed higher rates of lymph node invasion than other tumor types. The analysis of patients with gastric neoplasms is crucial for appropriate therapeutic management, and maintaining oncologic standards during medical crises is essential.

## 1. Introduction

Despite surgical advances, gastric neoplasia represents a major global health problem with significant incidence in many regions of the world, particularly in Asia and Latin America [1,2,3,4]. According to statistics from 2020, more than 1,089,000 new cases of gastric neoplasia and 769,000 associated deaths were reported worldwide [5,6,7,8].

Knowledge of histopathologic features is crucial for accurate diagnosis and prognosis determination [9,10]. Lauren’s histologic classification divides gastric adenocarcinomas into the intestinal type (associated with environmental factors like H. pylori, more common in men) and diffuse type (more frequent in women, associated with poorer prognosis) [10,11,12].

In 2019, the World Health Organization established the classification of gastric tumors. It provides a detailed framework for accurate diagnosis, dividing them into epithelial, mesenchymal, lymphoid, and neuroendocrine tumors. The most common subtype is gastric adenocarcinoma, further divided into the following variants: tubular adenocarcinoma, papillary adenocarcinoma, mucinous adenocarcinoma, pitted ring cell adenocarcinoma, and cribriform adenocarcinoma [1,13,14,15,16,17,18].

Gastric neoplasia treatment has evolved into a multimodal and personalized treatment approach [19,20]. While involving radiotherapy, chemotherapy, and immunotherapy, elective surgery with lymphadenectomy remains the only curative treatment [12,17,21,22].

The surgical interventions performed for malignant gastric tumors, depending on the anatomical location of the tumor (fundus, body, or antrum), are as follows: upper polar gastrectomy, total gastrectomy, and distal subtotal gastrectomy [21,22]. Lymphadenectomy is essential for staging and oncological outcomes and is classified as the D1, D2, or D3 type, with D2 lymphadenectomy combined with appropriate gastrectomy representing the “gold standard” in high-volume centers [23,24].

This article makes an original contribution to the existing literature by focusing on gastric neoplasia treatment during the COVID-19 pandemic. The COVID-19 pandemic has also put its “stamp” on the delay of this diagnosis, which brings with it a limitation on the number of curative interventions and an increase in the number of patients at a more advanced stage, disrupting health systems worldwide.

The aim of this article is to analyze the histopathological and surgical features observed in gastric neoplasia cases in a single surgical center from 2020 to 2021. This article aims to identify the predominant histological types and subtypes of gastric neoplasia, to highlight the relevant prognostic factors, to quantify the type of surgeries and the indices of curability (lymphadenectomy, resection margins), and to analyze the implications and possible correlations of these parameters as impact factors in the management of patients with gastric neoplasms.

## 2. Materials and Methods

This retrospective observational study was conducted at the Surgery II Clinic of “Pius Brînzeu” County Emergency Hospital, Timișoara. A total of 67 patients hospitalized between 1 January 2020 and 31 December 2021 with histopathologically confirmed gastric neoplasia were included. Data were extracted from the hospital’s InfoWorld electronic medical record system to identify associations between histopathologic parameters, surgical interventions, and patient prognosis.

Inclusion criteria were as follows: age ≥18 years, histopathologically confirmed gastric neoplasia, complete medical records, and patient consent.

Three data sources were analyzed in this paper: demographic information (age, sex), clinical records (diagnosis, surgical procedures), and histopathological findings (stage, grade, tumor characteristics). Histopathologic evaluation was based on the WHO 2019 classification criteria with the Lauren classification applied to adenocarcinomas. All slides were independently reviewed by two pathologists. Ki-67 assessment was performed in 6 cases, mainly GIST cases (n = 5), according to standard diagnostic protocols for risk stratification. Ki-67 assessment was limited due to institutional protocol constraints during the COVID-19 pandemic period.

The statistical analysis part was performed using SPSS Statistics version 29.0, with *p* < 0.05 considered significant. Descriptive statistics were used to assess normality using the Shapiro–Wilk test for continuous variables and frequencies/percentages for categorical variables. Spearman’s rank correlation coefficient was used for correlation analysis due to the non-normal distribution of data and sample size considerations.

Lymph node harvest variability varies based on clinical circumstances, including palliative interventions, local infiltration, and poor biological status, and depends on gastrectomy type, degree, patient status, and pandemic conditions.

This study was approved by the Ethics Committee of the County Emergency Hospital “Pius Brînzeu”, Timisoara. Patient data were coded to ensure anonymity and handled in accordance with GDPR regulations. All data uses respected patient privacy standards. The retrospective design introduces potential selection bias and relies on existing medical records, which may be incomplete. Non-randomized patient selection is a limitation in the study methodology.

This study has limitations, including its potential selection bias, incomplete medical records, small sample sizes, single-center design, missing Ki-67 data, lack of long-term follow-up outcomes, and high variability in lymph node harvest.

## 3. Results

### 3.1. General Characteristics of Study Population

This study included 67 patients with gastric neoplasia treated between January 2020 and December 2021 (Table 1). Adenocarcinoma was the predominant histological type (91.0%, n = 61), followed by GIST (7.5%, n = 5) and lymphoma (1.5%, n = 1). According to the Lauren classification, the intestinal type was the most common (49.2%), followed by the diffuse (36.1%) and mixed types (14.8%).

An analysis of continuous variables with respect to clinical characteristics (Table 2) revealed that advanced tumor stages were associated with increased lymph node involvement, with Stage III disease showing significantly more invaded nodes than early-stage tumors (*p* = 0.04). No significant differences were observed in age distribution or nodal sampling across Lauren classification subtypes (*p* = 0.73) or surgical approaches (*p* = 0.32), indicating consistent treatment approaches regardless of histological subtype.

Tumor staging, based on the TNM classification (AJCC, 8th edition), revealed a heterogeneous distribution [25]. As illustrated in Figure 1A, tumor staging revealed a concerning predominance of advanced disease, with Stage III being the most frequent (40.3%, n = 27), substantially exceeding Stage I presentations (20.9%, n = 14). This distribution suggests delayed diagnosis patterns, particularly notable during the pandemic period. Stage II represented 26.87% (n = 18), while Stage IV accounted for 11.94% (n = 8) of cases. Histological grading (Figure 1B) demonstrated aggressive tumor biology, with poorly differentiated (G3) tumors predominating at 53.7% (n = 36), more than double the rate of moderately differentiated tumors (G2, 26.87%, n = 18). Well-differentiated tumors (G1) were rare at 5.97% (n = 4), while undifferentiated (G4) and undetermined grade (Gx) represented 2.99% and 10.45%, respectively. Lymph node involvement was observed in 61.2% (n = 41) of patients, with a mean of 19.2 ± 12.8 lymph nodes examined (median 18, IQR 12–26, range 0–49) and a mean of 4.1 ± 6.7 lymph nodes invaded (median 3, IQR 0–6, range 0–38). Distant metastases (M1) were present in 11.94% (n = 8) of the cohort.

Lymphatic invasion (LV1) was identified in 58.2% (n = 39) of patients, vascular invasion (VI) in 10.4% (n = 7), and perineural invasion (PnI) in 58.2% (n = 39). The Ki67 proliferation marker was assessed in 9.0% (n = 6) of cases, with a median value of 45% (range: 10–70%), predominantly in GIST and advanced adenocarcinomas. These results are presented in Table 1.

The WHO classification revealed tubular adenocarcinoma as the most common (36%), followed by the mixed subtype (27%) and poorly cohesive type (13%). Next are the following subtypes: signet ring adenocarcinoma—11%; mucinous adenocarcinoma—7%; adenocarcinoma with special subtypes—4%; cribriform adenocarcinoma—2%; and papillary adenocarcinoma—2%.

Hematoxylin–eosin staining was used for the direct exposure of the histopathologic subtypes (according to the WHO classification) present in the studied group, as shown in optical microscopy images.

Representative histopathological images of different tumor types are shown in Figure 2, Figure 3, Figure 4, Figure 5, Figure 6 and Figure 7, illustrating the histological subtypes observed in our cohort.

### 3.2. The Curability Rate of Surgical Interventions in the Study Group

In the analyzed cohort, according to Figure 8, the surgical resection margins were negative (R0) in 88% of cases (n = 44), indicating complete tumor excision without residual microscopic invasion at the specimen margins. This result reflects an optimal surgical quality and is associated with better overall survival and a lower risk of local recurrence in the literature.

On the other hand, in 12% of cases (n = 6), positive margins (R1) were recorded, suggesting the presence of remaining tumor cells at the microscopic level, with unfavorable prognostic implications.

These cases may require additional adjuvant treatments and close oncological monitoring. The relatively low rate of R1 margins emphasizes the efficiency of the operative action in most procedures and may be an indirect indicator of good preoperative selection and correct surgical technique.

### 3.3. Analyzing the Batch According to the Type of Surgical Interventions

Of the 67 interventions performed, 50 (75%) were curative, and 17 (25%) were palliative. Palliative interventions were indicated in the presence of metastases, extensive tumor invasion with the impossibility of complete resection, or poor overall biological status.

The most common interventions were total subtotal distal gastrectomies (n = 29), followed by total gastrectomies (n = 11) and superior polar gastrectomies (n = 10).

The analyzed table highlights the distribution of the types of gastrectomy (subtotal, total, and upper polar) according to the degree of lymphadenectomy performed (D1, D2, D3), providing a clear perspective on current surgical practice.

Subtotal gastrectomy is predominantly performed with D2 lymph node dissection (19%) but also in a relevant proportion with D1 (8%) and, marginally, D3 (2%), suggesting that this intervention is preferred in cases where the tumor is distally located and the oncological stage allows for a conservative approach while maintaining the curative intent.

The distribution of resection margin status (R0/R1) across the three types of gastrectomy provides valuable insight into the oncological effectiveness and curability of different surgical approaches in gastric neoplasia treatment.

In patients who underwent subtotal distal gastrectomy, 28 cases of R0 resection (96.6%) and only 1 case of R1 resection (3.4%) were identified, giving this procedure the highest curative potential in the group studied. This suggests that, when correctly selected, this procedure can achieve complete resection in the majority of patients, especially when the tumor is located distal to the stomach. The low R1 resection rate further supports the suitability of this procedure for less invasive, early-stage gastric neoplasias, where a conservative approach combined with a D2 lymphadenectomy (as seen in many cases) can provide optimal results.

In the subgroup of patients in whom total gastrectomy was performed, an R0 rate of 72.7% and R1 rate of 27.3% indicate a relatively lower curative success compared to the subgroup in whom subtotal gastrectomy was performed. This result may be related to the higher complexity of the procedure, especially when tumors are located proximally or in advanced stages. The higher proportion of R1 resections suggests that some patients may not benefit fully from surgical resection alone, and adjuvant therapy may be required for these cases.

In the third subgroup, upper polar gastrectomy was practiced, which is accompanied by an R0 resection in 80% of cases (n = 8) and an R1 resection in 20% (n = 2). The slightly higher R1 rate suggests that obtaining clear margins in this region may be more difficult, and as with total gastrectomy, multimodality therapy may be an essential part of the treatment strategy.

The classification of surgical interventions according to type of lymphadenectomy and status of resectinal margins is illustrated in Figure 9.

The observed differences in R0 rates between surgical approaches should be interpreted cautiously given the wide confidence intervals, particularly for total gastrectomy (CI: 39.0–94.0%) and superior polar gastrectomy (CI: 44.4–97.5%), reflecting the limited statistical precision inherent in smaller subgroups.

### 3.4. Correlations Between Clinical and Pathological Variables

The relationship between continuous and ordinal clinical and pathological variables was examined using Spearman’s rank correlation coefficient (rho). Three specific correlations were analyzed to explore potential correlations relevant to disease progression and patient characteristics (Table 3).

First, a statistically significant positive correlation was observed between patient age and histological grade (rho = 0.28; *p* = 0.021). This finding suggests that older patients were more likely to present with higher-grade tumors (G3), indicating a potential correlation between age and tumor aggressiveness. Patients with poorly differentiated tumors (G3) had a mean age of 67.3 years, compared to 62.8 years for those with well- or moderately differentiated tumors (G1 or G2).

Second, according to Figure 10, a strong and highly significant positive correlation was identified between the number of lymph nodes examined and the number of lymph nodes invaded (rho = 0.65, *p* < 0.001; Figure 5). This result indicates that a greater extent of lymph node sampling was associated with an increased likelihood of detecting lymph node metastases, reflecting the extent of disease spread in surgically resected specimens. Among patients with more than 20 lymph nodes examined (n = 25), 76.0% (n = 19) had lymph node involvement, compared to 48.8% (n = 20) in those with fewer lymph nodes examined (n = 41).

Finally, in concordance with Table 4, the correlation between patient age and TNM stage was assessed but found to be non-significant (rho = 0.14; *p* = 0.27). This suggests that, in this cohort, age did not strongly influence the overall stage at diagnosis, with advanced stages (III and IV) distributed relatively evenly across age groups. For instance, Stage III was observed in 42.9% (n = 15) of patients under 65 years and 46.9% (n = 15) of those 65 years or older.

These findings highlight key relationships between clinical and pathological features, which are further summarized in Table 4.

### 3.5. Associations Between Tumor Characteristics and Invasive Features

The distribution of invasive features across key clinical subgroups is summarized in Figure 11. The associations between tumor characteristics and invasive features in this cohort was investigated using the Chi-square test. No significant association was found between patient age (categorized as <65 years vs. ≥65 years) and vascular invasion (χ^2^ = 0.92; df = 1; *p* = 0.34). Vascular invasion was observed in 62.9% (n = 22) of patients under 65 years and 53.1% (n = 17) of those 65 years or older, suggesting that age does not substantially influence this feature in our cohort. Similarly, the association between age and lymphatic invasion was not statistically significant (χ^2^ = 1.15; df = 1; *p* = 0.28), with lymphatic invasion present in 65.7% (n = 23) of younger patients and 50.0% (n = 16) of older patients.

According to Figure 11, Tumor type (adenocarcinoma [ADK], gastrointestinal stromal tumor [GIST], and lymphoma) showed a borderline association with vascular invasion (χ^2^ = 4.87; df = 2; *p* = 0.087). ADK patients exhibited vascular invasion in 59.0% (n = 36) of cases, compared to 40.0% (n = 2) in GIST and 100% (n = 1) in lymphoma; however, this trend did not reach statistical significance, likely due to the small number of GIST and lymphoma cases. In contrast, a significant association was observed between tumor type and lymphatic invasion (χ^2^ = 8.12; df = 2; *p* = 0.017). ADK patients had a higher prevalence of lymphatic invasion (63.9%, n = 39) compared to GIST (20.0%, n = 1), with the single lymphoma case showing no lymphatic invasion (0%). This suggests that adenocarcinomas are more likely to exhibit lymphatic spread than GISTs.

The association between tumor type and perineural invasion was not significant (χ^2^ = 3.45; df = 2; *p* = 0.18). Perineural invasion was present in 60.7% (n = 37) of ADK cases and 40.0% (n = 2) of GIST cases and absent in the lymphoma case, indicating no clear differential pattern across tumor types in this cohort.

In concordance with Figure 11, among adenocarcinomas, the Lauren classification (intestinal, diffuse, or mixed) was significantly associated with lymphatic invasion (χ^2^ = 6.23; df = 2; *p* = 0.044). Diffuse-type adenocarcinomas showed the highest rate of lymphatic invasion (77.3%, n = 17), followed by mixed-type (55.6%, n = 5) and intestinal-type (56.7%, n = 17), highlighting a greater propensity for lymphatic spread in the diffuse subtype.

These results, summarized in Table 5, provide insights into the invasive behavior of gastric neoplasia in relation to patient and tumor characteristics.

## 4. Discussion

### 4.1. Interpretation of Main Findings

Gastric neoplasia remains a formidable challenge that requires early detection and personalized therapeutic approaches based on multidisciplinary expertise. Current knowledge highlights the importance of understanding the histopathological forms of gastric neoplasia (the main types and subtypes and recognized classifications such as the Lauren classification), due to the desire to accurately categorize patients.

This single-center study during the COVID-19 pandemic provides important insights into gastric neoplasia histopathology and surgical outcomes. The present analysis identified three major tumor types, adenocarcinomas (91.0%), gastrointestinal stromal tumors (7.5%), and lymphomas (1.5%), with Stage III disease being the most common (40.3%) and poorly differentiated (G3) tumors predominating (53.7%). The most significant finding was the strong association between tumor type and lymphatic invasion (*p* = 0.017), with adenocarcinomas demonstrating much higher rates of lymphatic spread compared to GISTs, highlighting their more aggressive biological behavior.

### 4.2. Pandemic Context and Clinical Impact

This study, conducted during COVID-19, maintained high surgical standards with an 88% R0 resection rate despite pandemic challenges. Specifically, the surgical center where the patients were treated maintained high surgical standards, with an R0 resection rate of 88%. From a morphopathological point of view, there is a high incidence of the presentation of patients in Stage III, which can also be justified by the phenomenon of “stage delay” due to the pandemic.

### 4.3. Balanced Assessment of Clinical Outcomes

While maintaining an 88% R0 resection rate during the pandemic demonstrates surgical quality preservation, the predominance of Stage III disease (40.3%) versus Stage I (20.9%) represents a concerning failure in early detection. This “stage migration” significantly undermines the clinical impact of excellent surgical technique, as advanced-stage presentations inherently limit curative potential regardless of procedural success.

The high rates of lymph node involvement (61.2%) and lymphatic invasion (58.2%) reflect the consequences of delayed diagnosis that surgical excellence cannot compensate for. Our 25% palliative intervention rate illustrates cases where optimal surgical capability could not achieve cure due to late presentation.

These findings highlight that future quality improvement should prioritize early detection over further surgical refinement. Strategies should include enhanced screening protocols for high-risk populations, pandemic-resistant diagnostic pathways, and expedited referral systems for alarm symptoms.

### 4.4. Histopathological Validation

Our histopathological findings align with the established literature, with the adenocarcinoma distribution being consistent with Werner et al.’s findings [10].

The distribution of our patients according to the Lauren classification (intestinal 49.2%, diffuse 36.1%, mixed 14.8%) shows information about tumor biology and tumor invasion patterns. For example, diffuse adenocarcinomas had the highest rates of lymphatic invasion (77.3%), confirming their more aggressive character.

In our group, tumors with lower incidence such as GIST (7.5%) and gastric lymphoma (1.5%) were highlighted, which clearly required personalized therapeutic approaches. As highlighted in the current study, the low incidence of these tumors was evident even during the pandemic period. ESMO guidelines recommend immunohistochemical markers (CD-117, DOG1, CD34, SMA) for GIST characterization, highlighting the need for specialized management strategies [26].

### 4.5. Surgical Approach and D2 Lymphadenectomy

Concerning the type of gastrectomy performed and the associated lymphadenectomy, in the present study, the distribution of the types of gastrectomy (subtotal, total, and upper polar) according to the degree of lymphadenectomy performed was analyzed. Subtotal gastrectomy is predominantly performed with D2 lymph node dissection (19%), indicating compliance with current therapeutic standards. D2 lymphadenectomy is the gold standard in the treatment of gastric neoplasia. The majority of cases (66–91% depending on gastrectomy type) chose D2 lymphadenectomy, which brings superior benefits in terms of correct staging and a reduction in the risk of local recurrence.

One of the most representative studies in the literature, led by Songun et al., concludes that after a 15-year follow-up, D2 lymphadenectomies are associated with a lower loco-regional recurrence rate and lower gastric neoplasia mortality compared to D1 lymphadenectomies [27]. These results support the growing consensus that D2 lymphadenectomy represents the preferred standard in the surgical treatment of gastric neoplasia in specialized centers.

### 4.6. RO Index

An additional index of curability in this study is the assessment of the status of the resection margins in the study group and their correlation with the type of surgery performed. Specifically, distal subtotal gastrectomy demonstrates the highest curative potential, with 28 cases of R0 resection (96.6%) and only 1 case of R1 resection (3.4%). Total gastrectomy has a lower curative success rate (72.7%) and a higher R1 rate (27.3%), suggesting that some patients may not fully benefit from surgical resection alone. Upper polar gastrectomy has an intermediate level of curative success (80.0%) and a higher R1 rate (20.0%), suggesting the need for multimodal therapy.

The lower R1 index is a beneficial outcome, being associated with lower distant recurrence rates and lower morbidity and mortality, with the ultimate goal or ultimate aim being that all gastrectomies should be R0 in the future. Multiple studies in the literature report major benefits associated with RO-type resections.

For example, a study by Biondi et al. states that the direction of all therapeutic strategies in gastric neoplasia is to achieve a curative resection (RO), which is associated with significant improvement in the quality of life, decreased morbidity and mortality, and improved survival [28].

Two other specialty studies, one led by Ridwelski et al. and the other by Figueiredo et al., describing outcome data of R1 gastrectomized patients, show that an R1 situation worsens long-term survival, particularly for low T-stage and lymph node metastases [29,30].

Another study, led by Cordero-García et al., supports the impact of a set of parameters on improving survival and reducing recurrence. This set consists of the following: the status of resection margins, tumor type, and the degree of differentiation [31].

In this context, according to Figure 12, we propose naming a set of indices (resection margin status, type of lymphadenectomy performed, tumor type, degree of differentiation) as curability indices of gastric neoplasia and to present the results obtained for our group.

### 4.7. Adjuvant Therapy

The place of adjuvant therapy in gastric neoplasms remains a subject of ongoing debate in the literature, with significant geographical variations in treatment approaches [19,32,33,34,35,36,37].

Meta-analyses demonstrate that adjuvant fluorouracil-based chemotherapy provides a 6% absolute survival benefit compared to surgery alone (HR 0.82; 95% CI 0.76–0.90; *p* < 0.001) [38]. Recent studies on cytoreductive surgery with HIPEC show survival benefits particularly relevant for our cohort, given that 40.3% were Stage III with 58.2% lymphatic invasion [39].

### 4.8. Study Limitations

This study has several important limitations that must be acknowledged when interpreting the results. The relatively small sample size (n = 67) significantly limits statistical power for subgroup analyses, particularly for comparisons between rare tumor types such as GIST (n = 5) and lymphoma (n = 1) versus adenocarcinomas.

Additionally, the limited sample size precluded meaningful multivariable analysis to control for potential confounding factors such as age, tumor stage, and comorbidities simultaneously.

The retrospective observational design introduces inherent selection bias through the inclusion of only patients with complete medical documentation, potentially excluding cases representing different disease severity or management approaches, while information bias may arise from inconsistencies in data recording across different time periods and healthcare providers.

The study period (2020–2021) coinciding with the COVID-19 pandemic significantly impacts generalizability, likely contributing to delayed diagnoses, as evidenced by the high proportion of Stage III presentations (40.3%) and potential “stage migration” effects.

Resource limitations during the pandemic affected diagnostic comprehensiveness, including limited Ki-67 assessment and variations in lymph node harvesting techniques that may not reflect standard care capabilities.

Most critically, the absence of long-term follow-up data prevents the assessment of crucial oncological outcomes such as overall survival, disease-free survival, and recurrence patterns, limiting our ability to correlate the histopathological findings with clinical prognosis or validate surgical outcome classifications through real-world results.

As a single-center study conducted in Romania, the findings may not be generalizable to other healthcare systems, geographic regions, or patient populations with different characteristics, while the observational design precludes establishing causality between identified associations, representing correlations rather than causal relationships.

This study’s limitations include a small sample size in certain surgical subgroups, particularly total gastrectomy and superior polar gastrectomy, which limits the statistical power and reliability of comparisons, and the results should be interpreted cautiously due to wide confidence intervals and unstable estimates.

Despite these limitations presented by the pandemic context, our findings demonstrate the maintenance of surgical quality during the pandemic, with an R0 resection rate of 88%.

## 5. Conclusions

Gastric neoplasia is a neoplasm that continues to amaze researchers with its large number of new cases and systemic involvement and is often labeled as a global burden in terms of associated long-term care costs for these patients. This single-center study of 67 patients with gastric neoplasia during the COVID-19 pandemic presents important insights into histopathologic patterns and surgical outcomes.

This study on the histopathologic aspects of gastric neoplasia identified three major tumor types, adenocarcinomas (91.0%), gastrointestinal stromal tumors (GISTs, 7.5%), and lymphomas (1.5%), with adenocarcinomas being the most common. Stage III disease was the most frequent (40.3%), predominantly poorly differentiated (G3, 53.7%). The most significant finding was the association between tumor type and lymphatic invasion (*p* = 0.017), with adenocarcinomas showing a higher tendency for lymphatic dissemination compared to GIST and lymphomas, highlighting their more aggressive biological behavior.

From a surgical point of view, it can be concluded that curative interventions remain the most common (75%), with distal subtotal gastrectomy with D2 lymphadenectomy presenting the highest curative success (96.6% R0), emphasizing the importance of the type of surgery and the extent of lymphadenectomy. These findings have direct clinical implications: the strong association between adenocarcinomas and lymphatic invasion (*p* = 0.017) should guide staging protocols and treatment decisions, while the superior R0 rates obtained with subtotal gastrectomy plus D2 lymphadenectomy (96.6%) support this approach as the preferred one.

This study demonstrates the surgical quality maintained during pandemic conditions (88% R0 rate) while revealing concerning patterns of delayed presentation (40.3% Stage III). The findings provide a regional validation of international classification systems and confirm established associations between tumor characteristics and biological behavior. While technical surgical outcomes remained robust, the clinical impact is limited by late-stage presentations that surgical excellence cannot overcome.

Future improvements must prioritize early detection infrastructure alongside continued surgical standard maintenance. This study’s contributions are primarily confirmatory, highlighting the need for systematic approaches to earlier diagnosis in gastric neoplasia management.

## Figures and Tables

**Figure 1 cancers-17-02219-f001:**
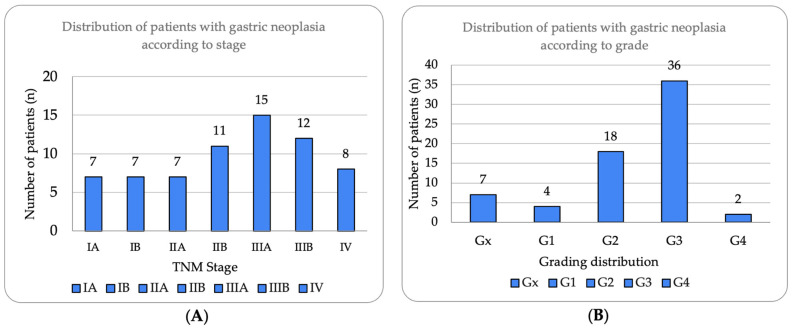
Distribution of patients with gastric neoplasia according to stage and histological grade. (**A**) Tumor stage distribution according to TNM classification (8th edition), showing Stage III as most prevalent (n = 27, 40.3%), followed by Stage II (n = 18, 26.87%), Stage I (n = 14, 20.90%), and Stage IV (n = 8, 11.94%). Substages IIIA and IIIB represented 22.39% and 17.91%, respectively. (**B**) Histological grade distribution demonstrating predominance of poorly differentiated tumors (G3, n = 36, 53.7%), followed by moderately differentiated (G2, n = 18, 26.87%), well-differentiated (G1, n = 4, 5.97%), undifferentiated (G4, n = 2, 2.99%), and undetermined grade (Gx, n = 7, 10.45%). High prevalence of G3 tumors correlates with advanced stage presentation.

**Figure 2 cancers-17-02219-f002:**
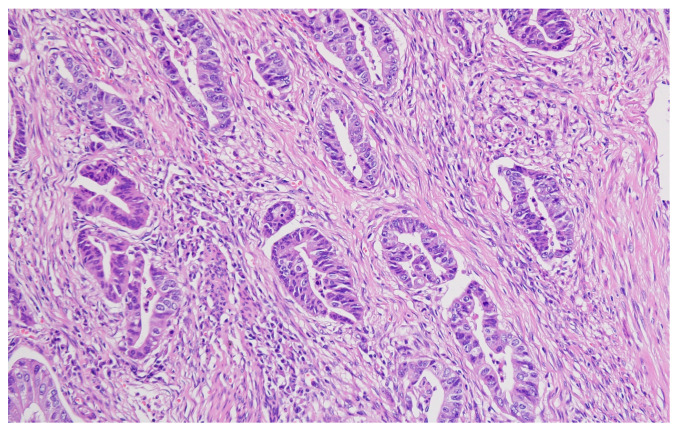
Tubular adenocarcinoma showing well-formed glandular structures with moderate nuclear atypia (H&E, ×200, scale bar = 100 μm).

**Figure 3 cancers-17-02219-f003:**
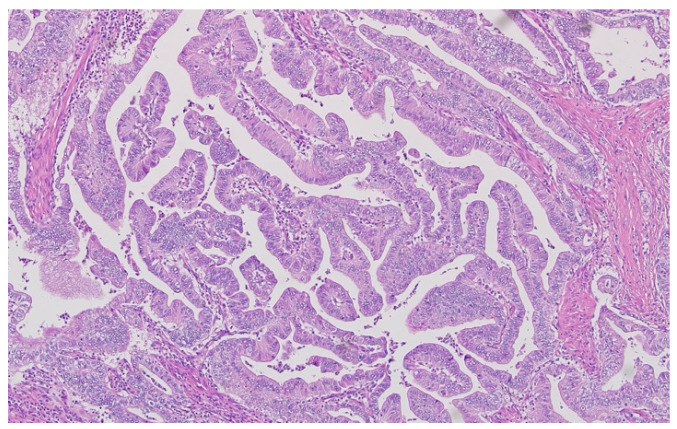
Poorly cohesive adenocarcinoma with signet ring cell morphology infiltrating the stroma (H&E, ×400, scale bar = 50 μm).

**Figure 4 cancers-17-02219-f004:**
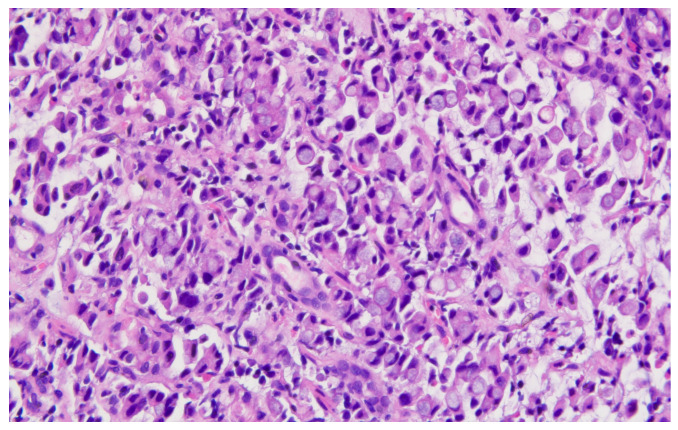
Mucinous adenocarcinoma with abundant extracellular mucin pools surrounding malignant glands (H&E, ×100, scale bar = 200 μm).

**Figure 5 cancers-17-02219-f005:**
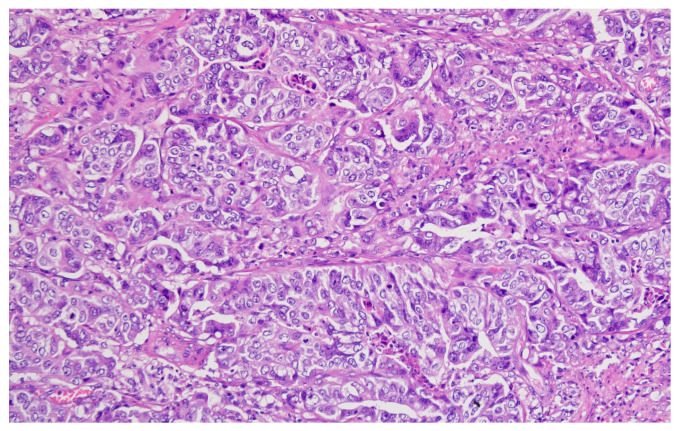
Moderately differentiated (G2) adenocarcinoma with irregular glandular architecture and moderate nuclear pleomorphism (H&E, ×200, scale bar = 100 μm).

**Figure 6 cancers-17-02219-f006:**
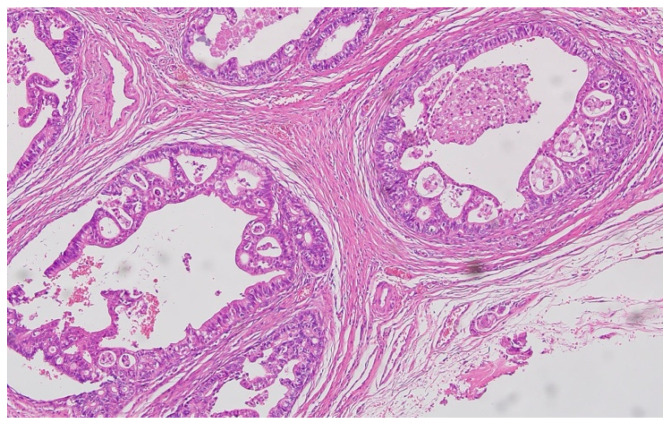
Poorly differentiated (G3) adenocarcinoma showing solid growth pattern with high nuclear pleomorphism and increased mitotic activity (H&E, ×200, scale bar = 100 μm).

**Figure 7 cancers-17-02219-f007:**
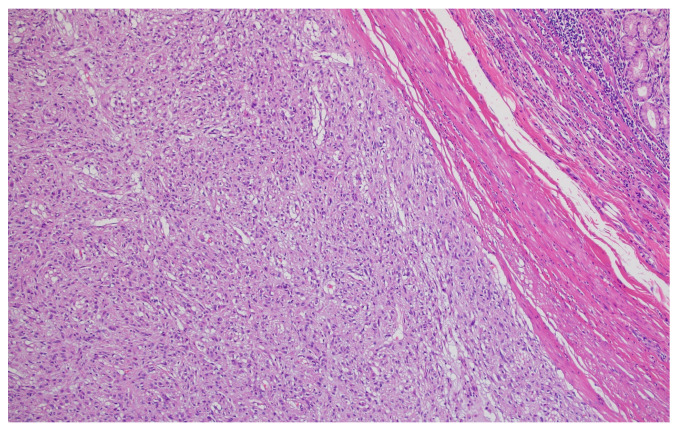
Lymphatic invasion demonstrating tumor cells within lymphatic vessels, supporting our finding of 58.2% lymphatic invasion rate (H&E, ×400, scale bar = 50 μm).

**Figure 8 cancers-17-02219-f008:**
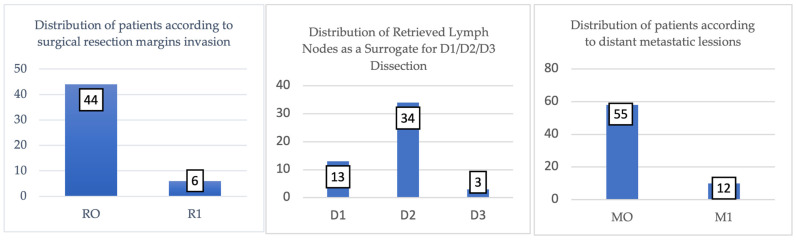
Surgical resection margin status in study cohort (n = 50 curative resections). R0 (negative margins) achieved in 44 cases (88%), indicating complete tumor excision without residual microscopic disease at specimen margins. R1 (positive microscopic margins) found in 6 cases (12%), suggesting remaining tumor cells with implications for adjuvant therapy and prognosis. Chart shows absolute numbers and percentages for each category.

**Figure 9 cancers-17-02219-f009:**
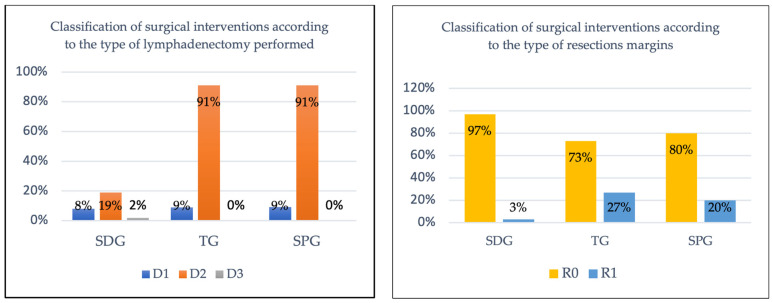
Classification of surgical interventions according to type of lymphadenectomy and status of resectional margins.

**Figure 10 cancers-17-02219-f010:**
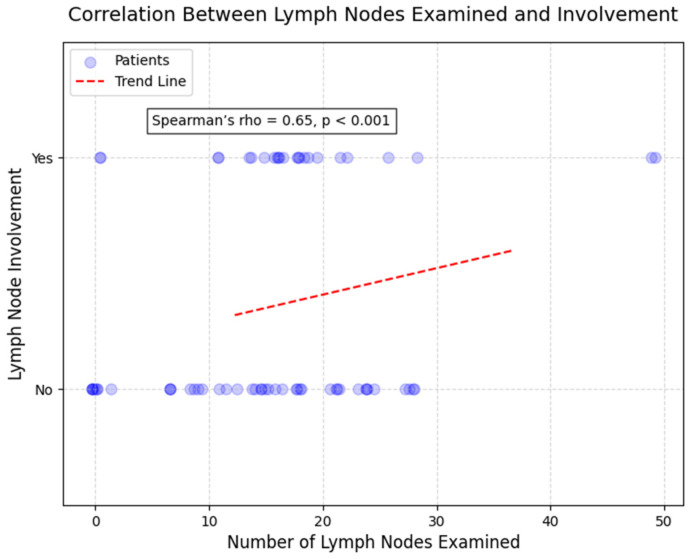
Scatterplot correlation between number of lymph nodes examined and number of lymph nodes invaded (Spearman’s rho = 0.65, *p* < 0.001). Each point represents one patient. Strong positive correlation demonstrates that more extensive lymph node sampling increases detection of nodal metastases. Patients with >20 nodes examined showed 76.0% lymph node involvement versus 48.8% in those with <20 nodes examined. Line represents best-fit trend.

**Figure 11 cancers-17-02219-f011:**
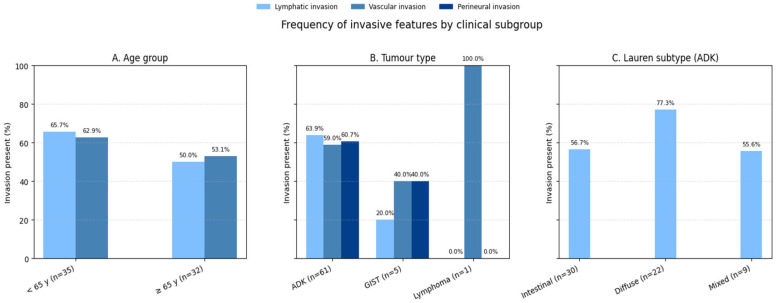
Distribution of invasive features across clinical subgroups. Bar chart comparing lymphatic invasion, vascular invasion, and perineural invasion rates between (**A**) age groups (<65 vs. ≥65 years), (**B**) tumor types (adenocarcinoma vs. GIST vs. lymphoma), and (**C**) Lauren classification subtypes (intestinal vs. diffuse vs. mixed). Error bars represent 95% confidence intervals. Adenocarcinomas showed significantly higher lymphatic invasion rates (63.9%) compared to GIST (20.0%, *p* = 0.017).

**Figure 12 cancers-17-02219-f012:**
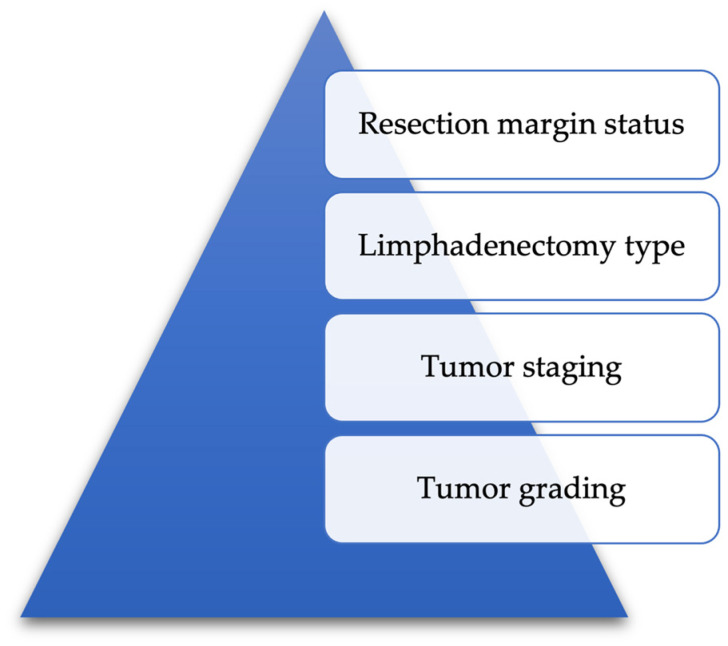
Establishing a set of curability indicators for gastric cancer surgery.

**Table 1 cancers-17-02219-t001:** General characteristics of study population.

Parameter	Value
**DEMOGRAPHIC CHARACTERISTICS**	
Age (years)	
Mean ± SD	65.7 ± 12.4
Median (IQR)	67 (58–74)
Range	30–82
Gender, n (%)	
Male	38 (56.7)
Female	29 (43.3)
**HISTOLOGICAL CHARACTERISTICS**	
Histological Type, n (%)	
Adenocarcinoma	61 (91.0)
GIST	5 (7.5)
Lymphoma	1 (1.5)
Lauren Classification (for ADK), n (%)	
Intestinal	30 (49.2)
Diffuse	22 (36.1)
Mixed	9 (14.8)
WHO Classification (for ADK), n (%)	
Tubular	22 (36.0)
Mixed	16 (27.0)
Poorly cohesive	8 (13.0)
Signet ring	7 (11.0)
Mucinous	4 (7.0)
Special subtypes	2 (4.0)
Cribriform	1 (2.0)
Papillary	1 (2.0)
**TUMOR STAGING AND GRADING**	
Tumor Stage (TNM), n (%)	
Stage I	14 (20.90)
Stage II	18 (26.87)
Stage III	27 (40.30)
—IIIA	15 (22.39)
—IIIB	12 (17.91)
Stage IV	8 (11.94)
Histological Grade, n (%)	
G1 (Well-differentiated)	4 (5.97)
G2 (Moderately differentiated)	18 (26.87)
G3 (Poorly differentiated)	36 (53.73)
G4 (Undifferentiated)	2 (2.99)
Gx (Undetermined)	7 (10.45)
LYMPH NODE ASSESSMENT	
Lymph Nodes Examined	
Mean ± SD	19.2 ± 12.8
Median (IQR)	18 (12–26)
Range **	0–49
Lymph Nodes Invaded	
Mean ± SD	4.1 ± 6.7
Median (IQR)	3 (0–6)
Range	0–38
Lymph Node Involvement, n (%)	41 (61.2)
**INVASION PATTERNS**	
Distant Metastasis (M1), n (%)	8 (11.94)
Lymphatic Invasion (LV1), n (%)	39 (58.2)
Vascular Invasion (VI), n (%)	7 (10.4)
Perineural Invasion (PnI), n (%)	39 (58.2)
**PROLIFERATION MARKER**	
Ki67 * Assessment, n (%)	6 (9.0)
Ki67 Proliferation Index (n = 6)	
Mean ± SD	42.5 ± 22.3%
Median (IQR)	45 (25–60)%
Range	10–70%

* Ki-67 was evaluated primarily in GIST cases (n = 5) according to standard diagnostic protocols and in selected advanced adenocarcinomas, as per the institutional protocol during the pandemic period. ** Cases with 0 lymph nodes examined included patients undergoing palliative procedures where lymphadenectomy was not performed due to advanced disease stage, extensive local invasion, or poor performance status.

**Table 2 cancers-17-02219-t002:** Comparison of continuous variables in terms of clinical characteristics.

	Age (Years)	Nodes Examined	Nodes Invaded	
	Mean ± SD	Mean ± SD	Mean ± SD	*p*-value
Tumor Stage				
Stage I (n = 14)	62.1 ± 11.8	22.4 ± 8.9	1.2 ± 2.1	
Stage II (n = 18)	64.8 ± 13.2	20.1 ± 12.4	3.8 ± 4.2	
Stage III (n = 27)	68.2 ± 11.9	18.7 ± 14.2	6.2 ± 8.9	0.04 *
Stage IV (n = 8)	67.9 ± 14.1	14.2 ± 11.8	4.9 ± 6.1	
Lauren Classification (ADK only)				
Intestinal (n = 30)	67.8 ± 12.1	20.4 ± 11.9	3.9 ± 5.8	
Diffuse (n = 22)	64.2 ± 11.8	18.1 ± 13.8	4.8 ± 7.2	0.73
Mixed (n = 9)	63.1 ± 14.2	17.8 ± 12.1	3.2 ± 6.1	
Surgery Type				
Subtotal (n = 29)	65.1 ± 13.2	21.3 ± 11.7	3.8 ± 5.9	
Total (n = 11)	68.4 ± 10.8	16.2 ± 15.3	5.2 ± 8.4	0.32
Polar (n = 10)	63.8 ± 12.9	18.9 ± 13.1	4.1 ± 7.2	

* Statistically significant (*p* < 0.05).

**Table 3 cancers-17-02219-t003:** Evaluating the oncological effectiveness of surgical procedures based on lymphadenectomy level (D1, D2, D3) and resection status (R0, R1).

SURGICAL INTERVENTIONS		SUBTOTAL DISTAL GASTRECTOMY (n = 29)	TOTAL GASTRECTOMY (n = 11)	SUPERIOR POLAR GASTRECTOMY (n = 10)
**LYMPHADENECTOMY TYPE**	D1 (stations 1–6)	8 (28%)	1 (9%)	4 (10%)
	D2 (D1 + stations 7–12)	19 (66%)	10 (91%)	5 (50%)
	D3 (D2 + stations 16)	2 (7%)	0 (0%)	1 (10%)
**RESECTION MARGIN STATUS**				
	R0	28 (97%) [CI: 82.8–99.4%] *	8 (73%) [CI: 39.0–94.0%]	8 (80%) [CI: 44.4–97.5%]
	R1	1 (3%) [CI: 0.6–17.2%]	3 (27%) [CI: 6.0–61.0%]	2 (20%) [CI: 2.5–55.6%]

D1 includes only perigastric nodes (stations 1–6) located along the lesser and greater curvatures and around the pylorus. D2 extends dissection to second-tier nodes along major arteries, including the left gastric artery (station 7), common hepatic artery (station 8a), celiac trunk (station 9), splenic hilum (station 10), splenic artery (stations 11p/11d), and the proper hepatic artery within the hepatoduodenal ligament (station 12a). D3 further includes para-aortic nodes (station 16). * CI = confidence interval (95%). Wide confidence intervals in smaller subgroups (total and polar gastrectomy) reflect limited statistical precision due to sample size constraints.

**Table 4 cancers-17-02219-t004:** Spearman correlation coefficients for clinical and pathological variables.

Variable Pair	Spearman’s Rho	*p*-Value
Age vs. Histological Grade (G)	0.28	0.021
Lymph Nodes Examined vs. Invaded	0.65	<0.001
Age vs. TNM Stage	0.14	0.27

**Table 5 cancers-17-02219-t005:** Associations between tumor characteristics and invasive features.

Variable Pair	χ^2^ Value	Degrees of Freedom (df)	*p*-Value
Age (< 65 vs. ≥65) vs. Vascular Invasion (LV)	0.92	1	0.34
Age (<65 vs. ≥65) vs. Lymphatic Invasion (LV)	1.15	1	0.28
Tumor Type vs. Vascular Invasion (LV)	4.87	2	0.087
Tumor Type vs. Lymphatic Invasion (LV)	8.12	2	0.017
Tumor Type vs. Perineural Invasion (Pn1)	3.45	2	0.18
Lauren Classification vs. Lymphatic Invasion (LV)	6.23	2	0.044

## Data Availability

The original contributions presented in this study are included in the article; further inquiries can be directed to the corresponding author.

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
