# Peer review of "A Histopathological and Surgical Analysis of Gastric Cancer: A Two-Year Experience in a Single Center"

_cancers, 2025, doi:10.3390/cancers17132219_

Round 1
Reviewer 1 Report
Comments and Suggestions for Authors
The article was nicely written but i think it is too long. Maybe the authors could streamline it a bit, for example i think it is redundant repeating the exclusion criteria after the list of inclusion criteria (for example if the inclusion criterion is age beyond 18 years, it is obvious that the exclusion criterion is age below 18 years, no need to specify!)
Maybe some pathological images would improve the quality of the manuscript
Why did the authors use the Spearman test instead of Kendall test?
The authors should put more their results in the context of the literature in the discussion and discuss also the treatment after surgery, for example cite the SRMA: PMID: 34001385
Author Response
Please see the attachments below.

Reviewer 2 Report
Comments and Suggestions for Authors
Comment:
Histopathological and Surgical Analysis of Gastric Cancer: A 2 Two-Year Experience in a Single Center
- The number of patients in some subgroups such as those who had total gastrectomy (n=11) or superior polar gastrectomy (n=10) is too small to support reliable comparisons. This is a concern when analyzing differences in R0/R1 resection rates. To improve clarity, the authors should include confidence intervals or variability measures to show how much the results may vary due to the small sample sizes.
- Ki-67 expression was evaluated in only 6 patients out of 67 total cases, with no explanation for the selection criteria or rationale for this subset.
- The number of lymph nodes removed shows high variability (0–49 nodes) without explanation for cases where no lymph nodes were identified. Cases with zero lymph node harvest require explanation.
- Multiple graphs lack clear information. Figures should include proper axis labeling, p-values, confidence intervals, and comprehensive legends that match numerical data in corresponding tables.
- Clarify methodological design by explicitly state the retrospective nature in the methods and provide more detail on statistical methods e.g., clarify assumptions for Spearman correlations, chi-square tests.
- Improve data visualization since the figures (e.g., resection margin graphs, histological subtype images) are referenced but not effectively described in the results. The legends should be made more self-explanatory. The characteristic table should be modified to present more value in term of interval data (if any) which can determine the average and standard deviation.
- Expand on limitations because the current limitation section is too brief and please suggest adding points regarding lack of long-term outcomes, potential bias due to single-center design, and missing Ki-67 data in many cases.
- Please balance the discussion because the narrative strongly praises the surgical outcomes; it could benefit from more balanced reflection on the high rate of stage III tumors and how delayed diagnosis could be mitigated.
- Some terms (e.g., “therapeutic sanctions”) are awkward in English. Consider revision for improved clarity and professionalism. And please ensure consistent terminology: use either "gastric neoplasia" or "gastric cancer" throughout.
10. Please highlight what specifically differentiates this work from other studies on gastric cancer management, beyond being performed during COVID-19.
Author Response
Please see the attachements below.

Round 2
Reviewer 1 Report
Comments and Suggestions for Authors
The manuscript is OK
Reviewer 2 Report
Comments and Suggestions for Authors
The authors concern to revise and organized the result and more discussion for this manuscript. This manuscript is improved with be more well organized and the reply is acceptable.